# Structural Optimization and Performance Evaluation of Blocking Wheel-Type Screw Fertilizer Distributor

**Xiantao Zha** [1,2], **Guozhong Zhang** [1,2,*], **Yuhang Han** [1,2], **Abouelnadar Elsayed Salem** [1,3], **Jianwei Fu** [1,2] **and Yong Zhou** [1,2]

1    College of Engineering, Huazhong Agricultural University, Wuhan 430070, China; zhaxiantao@webmail.hzau.edu.cn (X.Z.); hanyh@webmail.hzau.edu.cn (Y.H.); abouelnadar@webmail.hzau.edu.cn (A.E.S.); fjwtap@mail.hzau.edu.cn (J.F.); zhyong@mail.hzau.edu.cn (Y.Z.)
2    Key Laboratory of Agricultural Equipment in Mid-Lower Yangtze River, Ministry of Agriculture and Rural Affairs, Wuhan 430070, China
3    Soil Conservation Department, Desert Research Center, Matraia 11753, Egypt
*    Correspondence: zhanggz@mail.hzau.edu.cn

**Abstract:** In order to solve the problem where the amount of screw fertilizer distributor can only be adjusted by rotating speed and poor fertilization uniformity at low rotational speeds, a blocking wheel-type screw fertilizer distributor was designed. Single factor and $L_9(3^4)$ orthogonal simulation tests based on EDEM software were carried out to optimize the distributor variables at a speed of 20 r/min. The bench verification test was built under the same conditions as the simulation tests to verify the results of the simulation. Finally, the bench performance tests were carried out to evaluate distributor performance. The results of simulation tests revealed that the minimum coefficient of variation of fertilization uniformity (CVFU) was 19.27%, with the structural parameter combination of the inner diameter (17 mm), pitch (45 mm), outlet distance (40 mm), and number of screw heads (1). The verification test results showed that the changing trend and values of the CVFU were almost the same as the simulation tests. The results of the performance test revealed that when the opening width of the blocking wheel was 10–30 mm and the rotation speed was 20–60 r/min, the amount of fertilizer per lap (FAPL) was in the range of 27.74–38.15 g/r; the maximum CVFU and the coefficient of variation of fertilization stability (CVFS) were 29.43% and 2.18%, respectively, which met the requirements of the industry standard. This research provides a good reference for optimizing the screw fertilizer distribution and for researchers in the field of precision fertilization.

**Keywords:** screw fertilizer distributor; blocking wheel; discrete element method (DEM); fertilizing performance; fertilization uniformity

## 1. Introduction

Mechanized precision fertilization is an effective way to improve the fertilizer utilization rate and increase cost savings and efficiency during farming [1–4]. In order to solve the impact of the variable forward speed of machines on fertilization accuracy, researchers have developed machines using rotary encoders and Global Position System (GPS) to measure the forward speed of the machines and adjust the fertilizer quantity rate to the forward speed [5–7]. To meet the requirements of the different fertilizer amounts caused by different soil fertilities, Meng et al. [8,9] and Jeong et al. [10] carried out a map-based variable rate fertilizer application system. They used GPS to obtain the position information and ground wheels to obtain the forward speed of the machine. Then, the fertilization rate would be adjusted according to the forward speed, position information, and pre-stored prescription map. Kim et al. [11], Chen et al. [12], and Cho et al. [13] used sensors to obtain crop growth status or soil organic matter information to determine the fertilizer requirement. The fertilization rate was then be adjusted according to the fertilization requirements and the forward speed of the machine. To improve the fertilizer discharging precision

of the fertilizer distributor, Yuan et al. [14] used a controller based on Advanced RISC Machines/Digital Signal Processor (ARM/DSP) to control the electric push rod to change the effective working length of the groove wheel fertilizer distributor and to control the speed of the direct-current (DC) motor to adjust the speed of the screw shaft. The research referred to above is primarily concerned with speed control of the fertilizer distributor and coordination with forward speed and other decision-making information. However, the distributors were the basic components of the machines, which also had a significant impact on the performance of the machines.

In recent years, researchers have continued to innovate and optimize the structure of fertilizer distributor to improve their fertilization performance [15–18]. For example, to improve the stability and uniformity of fertilizer distributor, researchers optimized the structure of groove wheel [19], fertilizer tongue [20], and fertilizer guide devices [21]. Nevertheless, this type of distributor was only suitable for granular fertilizers with good fluidity and dryness, so its application has strong limitations. Because of its strong adaptability to fertilizers with different fluidity, the screw fertilizer distributor has received more attention. Nukeshev et al. [22] added a segment reflector and determined the screw diameter, the number of screw heads, and the number of ripples of reflector to screw distributor to solve the problem of unevenly discharging fertilizers. Chen et al. [23] used different kinds of granular fertilizer and powder fertilizer to evaluate the performance of double-level horizontal screw fertilizer distributor, which confirmed that the screw fertilizer distributor had high fertilization stability and adaptability to fertilizers with different fluidity. Cao [24] optimized the structural parameters of screw fertilizer distributor and analyzed the influence of screw speed on fertilizer distributing performance by DEM and bench test, determining the optimal screw structure parameters and working speed. Dong [25] studied the movement law of fertilizer in the vertical screw fertilizer distributor utilizing theoretical analysis, numerical simulation, and tests, determined its optimal structural parameters and working parameters, and proved that the vertical screw fertilizer distributor had good fertilization stability and uniform rotating speed. In particular, the above research examined the adaptability of the structure and working parameters of the screw fertilizer distributor of different fertilizers. The analysis of the influence of law and mechanism on the structure and working parameters of the screw fertilizer distributor on the fertilization uniformity was not sufficient. The fertilizer amount of existing screw fertilizer distributors can be only adjusted by the screw shaft rotation speed, and the uniformity of fertilizer discharge is poor when the rotation speed is low, so it is difficult to meet the requirements of precision fertilizer application.

Regarding the above-mentioned problems, this paper has designed a blocking wheel-type screw fertilizer distributor. (1) Structural parameters of the distributor were optimized by single factor and $L_9(3^4)$ orthogonal simulation tests at a low speed of 20 r/min based on EDEM software. (2) Bench verification tests under the same conditions as the simulation tests were carried out to verify the simulation results. (3) The performance of the distributor was evaluated by bench performance tests. This research provides a useful reference for the optimization of the screw fertilizer distributor and the research of precision fertilization technology.

## 2. Materials and Methods

### 2.1. Structural Design

The structure of the blocking wheel-type screw fertilizer distributor is shown in Figure 1. It consisted of a screw fertilizer transporting axis, a sleeve of screw, a pyramidal feeding inlet, a spiral blocking wheel, and a fertilizer outlet. In the figure, $D$ is the outer diameter of the screw blade, $d$ is the inner diameter of the screw blade, $P_t$ is the pitch, $b$ is the average thickness of the screw blade, $K$ is the spiral blocking wheel opening width, and $S$ is the outlet distance (short for distance between the end of screw blade and outlet).

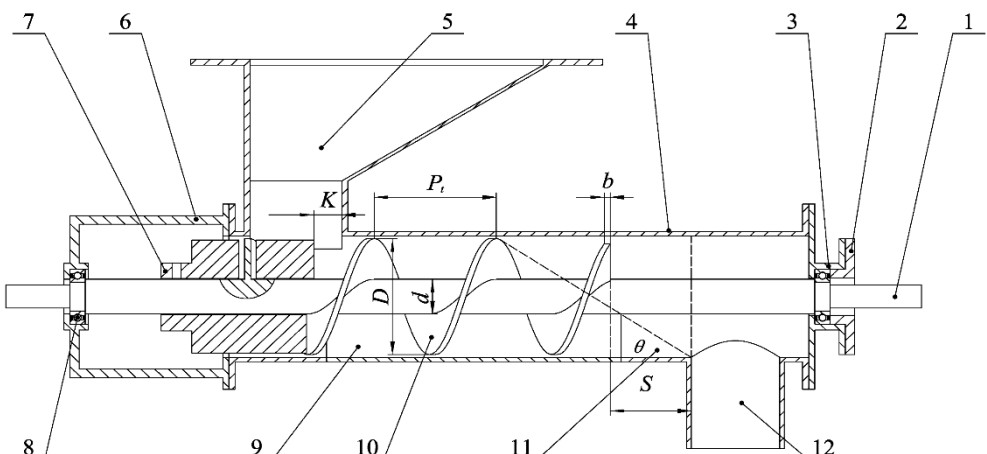

**Figure 1.** Structure of blocking wheel-type screw fertilizer distributor. 1. Screw shaft 2. Bearing cover 3. End cover of sleeve 4. Sleeve of screw 5. Pyramidal feeding inlet 6. Bearing seat 7. Spiral blocking wheel 8. Bearing 9. Fertilizer cleaning outlet and its cover 10. Screw blade 11. Fertilizer buffering zone 12. Fertilizer outlet. Note: $D$ is the outer diameter of the screw blade, mm; $d$ is the inner diameter of the screw blade, mm; $P_t$ is the pitch, mm; $b$ is the average thickness of the screw blade, mm; $K$ is the spiral blocking wheel opening width, mm; $S$ is the outlet distance, mm.

### 2.2. Operation Principle

When working, the fertilizer particles fill the beginning screw section through the feeding inlet under the action of their own gravity. With the rotation of the screw shaft, fertilizers in the beginning section of the screw are moved to the end of the screw under the action of the screw blade. Then they are discharged from the fertilizer outlet through the fertilizer buffer zone between the end of the screw blade and the fertilizer outlet.

Compared with the existing screw fertilizer distributor, a fertilizer buffer zone is set between the end of the screw blade and the fertilizer outlet, so that the fertilizer transported to the end of the distributor can be slowly discharged from the fertilizer outlet, which can improve the fertilization uniformity. The blocking wheel can be used to change the fertilizer amount per lap (FAPL) of the screw fertilizer distributor. As a result of the low target fertilization rate, the problem of low uniformity of fertilizer distributor can be resolved by reducing the fertilizer amount per lap and increasing the working speed of the screw shaft.

### 2.3. Force Analysis of Fertilizer Particles

The fertilizer particles in the fertilizer distributor are subjected to the combined effect of gravity $G$, the friction force $f_w$ generated by the outer shell wall of the fertilizer distributor, and the force of the screw blade on the fertilizer particles.

The force of the screw blade on the fertilizer particles is shown in Figure 2; the particles are subjected to the tangential friction force $f_s$ and the normal pressure $F_N$. Their resultant force is $F$. Force $F$ can be decomposed into an axial force $P_1$ and a circumferential force $P_2$. The axial force $P_1$ makes the fertilizer move to the outlet axially, and the circumferential force $P_2$ makes the fertilizer particles move in a circumferential direction. However, gravity $G$ and friction force $f_w$ overcomes this movement to make the fertilizer particles slide axially in the fertilizer distributor.

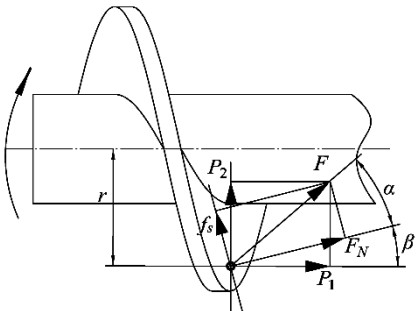

**Figure 2.** Force of screw blade on fertilizer particles. Note: $F$ is the total force of screw blade acting on fertilizer particles, N; $F_N$ is the normal pressure of screw blade on fertilizer particles, N; $f_s$ is the friction force of screw blade on fertilizer particles, N; $P_1$ is the axial component of $F$, N; $P_2$ is the circumferential component of $F$, N; $\alpha$ is the friction angle between blade and fertilizer, (°); $\beta$ is the lift angle of screw blade, (°); $r$ is the distance between particles and axis of screw shaft, mm.

According to Figure 2, the force $F$ is decomposed by Equation (1):

$$\begin{cases} P_1 = F\cos(\alpha + \beta) \\ P_2 = F\sin(\alpha + \beta) \\ \alpha = \tan^{-1}\mu \\ \beta = \tan^{-1}\frac{P_t}{2\pi r} \end{cases} \tag{1}$$

where $F$ is the total force of screw blade acting on fertilizer particles, N; $P_1$ is the axial component of $F$, N; $P_2$ is the circumferential component of $F$, N; $\alpha$ is the friction angle between blade and fertilizer, (°); $\beta$ is the lift angle of screw blade, (°); $\mu$ is the friction coefficient between fertilizer particles and screw blade; $P_t$ is the pitch of screw blade, mm; $r$ is the distance between particles and axis of screw shaft, mm.

*2.4. Key Performance Parameters of Screw Fertilizer Distributor*

2.4.1. Fertilizer Amount Per Lap (FAPL)

The FAPL is an important parameter to scale the fertilizer discharge capacity of the fertilizer distributor. It is generally believed that under ideal conditions, fertilizer does not move in a circle with the fertilizer screw, but only moves horizontally in the axial direction. Then the FAPL can be calculated by Equation (2) [21]:

$$q = \left[\pi\left(D^2 - d^2\right)P_t/4 - ZbHL_p\right]\rho\varphi \times 10^{-3} \tag{2}$$

where $q$ is FAPL, g; $D$ is outer diameter of screw blade, mm; $d$ is inner diameter of screw blade, mm; $P_t$ is pitch of screw blade, mm; $Z$ is the number of screw heads; $b$ is the average thickness of screw teeth, mm; $H$ is the depth of screw teeth, $H = \frac{D-d}{2}$, mm; $L_p$ is the average screw length of single circle, $L_p = \sqrt{\left[\frac{\pi(D+d)}{2}\right]^2 + t^2}$, mm; $\rho$ is the density of fertilizer, g/cm$^3$; $\varphi$ is the fertilizer filling rate of the fertilizer screw.

According to Equation (2), the main parameters affecting FAPL of the screw fertilizer distributor are the outer diameter of screw blade $D$, inner diameter of screw blade $d$, pitch $P_t$, and filling rate of the screw $\varphi$. Among them, the filling rate $\varphi$ is the ratio of the volume of fertilizer pushed by the blade to the volume formed by the blade for each revolution of the screw, which is mainly related to the material characteristics of fertilizer (such as particle shape, particle size, particle size distribution, and fluidity), blocking wheel opening width, vibration of the distributor, rotation speed of screw shaft, and other factors. When designing the structure parameters of the screw distributor according to FAPL, the outer diameter of the screw blade $D$, inner diameter of the screw blade $d$, and pitch of the blade $P_t$ should be mainly considered.

In actual operation, when the fertilizer exhauster works stably, the equation of FAPL is as follows:

$$q = \frac{60 \cdot \Delta m}{n \cdot \Delta t} \tag{3}$$

where $q$ is FAPL, g/r; $\Delta t$ is the time interval between the beginning and the end of a measurement, s; $\Delta m$ is the total mass of discharged fertilizer during the period of $\Delta t$, g; $n$ is the rotation speed of fertilizer screw, r/min.

2.4.2. Coefficient of Variation of Fertilization Uniformity and Stability (CVFU and CVFS)

CVFU and CVFU are important indices for assessing the uniformity and stability of the fertilizer distributor. Their test method in this research followed the China Industry Standard NY/T 1003-2006 (Technical specification of quality evaluation for fertilization machinery) [26].

The CVFU should be tested according to the following steps. First, ensure that the fertilization machine passes through a working length of more than 10 m at normal operating speed smoothly. Then take a length of 3 m from it, where the machine is stably forward. Thirdly, divide the 3 m length of fertilizers into 30 sections continuously. Fourth, the fertilizer quantity of each section should be weighed with an electronic balance (measurement accuracy is not less than 0.1 g). Finally, the CVFU could be calculated by Equation (4):

$$\begin{cases} m = \frac{1}{a} \sum\limits_{i=1}^{a} m_i \\ S = \sqrt{\frac{1}{a-1} \sum\limits_{i=1}^{a} (m_i - m)^2} \\ V = \frac{S}{m} \times 100 \end{cases} \tag{4}$$

where $m_i$ is the quantity of fertilizer in the $i$-th section ($i = 1, 2, 3, \ldots, 30$), g; $m$ is the average value of fertilizer quantity of the 30 section, g; $a$ is the total number of the sections, which is equal to 30 when measuring the CVFU; $S$ is the standard deviation of fertilizer quantity, g; $V$ is the CVFU, %.

The CVFS is also calculated by Equation (4), but the testing method is not the same. It should be tested as follows. Firstly, take the fertilizer distributed by the distributor within a period (not less than 2 min). Secondly, weigh the fertilizer quantity with an electronic balance (measurement accuracy is not less than 0.5 g). Thirdly, repeat the above two steps five times, and each test should last for a same period. Then, $m_i$ is the quantity of fertilizer in the $i$-th test ($i = 1, 2, 3, 4, 5$), g; $m$ is the average value of fertilizer quantity of the 5 tests, g; $a$ is the total number of the tests, which is set to 5 when measuring the CVFS; $S$ is the standard deviation of fertilizer quantity, g; $V$ is the CVFS, %.

The equation for the calculation of CVFU and CVFS is the same. However, according to the standard (NY/T 1003–2006), the number of samples required for calculating CVFU is 30, and the fertilization distance of each sample is 0.1 m. Thus, CVFU reflects the uniformity and stability of the fertilization quantity over a short time and distance. On the contrary, the number of samples required for calculating the CVFS is 5, and the measurement time of each sample is not less than 2 min (if the machine moves forward at a speed of 0.5 m/s, the fertilization distance is 60 m within 2 min). The CVFS therefore reflects the uniformity and stability of the fertilization quantity over a long time and distance.

*2.5. Methods of DEM Simulation Tests*

In order to facilitate the research, this paper has fixed the outer diameter of the screw blade to $D = 45$ mm and analyzed the influence of the parameters (inner diameter of the screw blade $d$, pitch $P_t$, number of screw heads $Z$, outlet distance $S$, and the opening width of the locking wheel $K$) on FAPL and CVFU. All the simulations were carried out at a low screw shaft rotation speed of 20 r/min.

### 2.5.1. Simulation Model and Parameter Setting

Studies have shown that the particles of compound fertilizer are mostly spherical or ellipsoidal discrete elements, with high sphericity and no adhesion between the particles [27–29], so this paper chose the "Hertz-Mindlin (no slip)" contact model to simulate the composite fertilizer particles. Referring to the existing research, the average particle size fertilizer particles $d_s$ was set to 3.3 mm, the generation method of the particle size range as "random", with a setting range of 0.75–1.25. Then, the diameter of generated particles was 2.475–4.125 mm [28,29]. The other simulation parameters are shown in Table 1.

**Table 1.** Simulation parameters.

| Parameters | Value |
| --- | --- |
| Poisson's ratio of fertilizer | 0.28 |
| Density of fertilizer (kg/m$^3$) | 1511 |
| Shear modulus of fertilizer (Pa) | $1 \times 10^7$ |
| Poisson's ratio of ABS | 0.394 |
| Density of ABS (kg/m$^3$) | 1060 |
| Shear modulus of ABS (Pa) | $8.9 \times 10^8$ |
| Coefficient of Restitution between fertilizers | 0.35 |
| Coefficient of Static Friction between fertilizers | 0.4774 |
| Coefficient of Rolling friction between fertilizers | 0.21 |
| Coefficient of restitution between fertilizer and ABS | 0.359 |
| Coefficient of Static Friction between fertilizer and ABS | 0.1962 |
| Coefficient of restitution between fertilizer and ABS | 0.15 |

### 2.5.2. Calculation of CVFU in Simulation Tests

The test method in Section 2.4.2 made the simulation domain too large, which led to low efficiency of the simulation test. To increase the simulation speed, this paper carried out the simulation test according to the following method.

In the preparation step, a 3D model of the fertilizer distributor with a needed blocking wheel width *K* was built in Solidworks software.

During the simulation model set-up step, the 3D model was imported into EDEM and the rotation speed of screw shaft was set to 20 r/min. Other simulation parameters were set referring to Table 1. Then a particle factory was built to add 1 kg of fertilizer particles to the fertilizer box, and all the particles were generated within 1 s. The fertilizer screw was rotated after the particles in the fertilizer box were steady.

After the simulation was done, the FAPL of the screw fertilizer distributor could be calculated as follows. Firstly, a total mass sensor for remained fertilizer was built as shown in Figure 3. Secondly, after the fertilizer had been distributed steadily, the timeline was two different times and two total mass values and were obtained by the sensor, then the FAPL could be calculated with Equation (3).

The CVFU of the screw fertilizer distributor could be calculated as follows. Firstly, a cylindrical mass sensor for the fertilizers was built at the fertilizer outlet as shown in Figure 3. Secondly, the mass of the fertilizer discharged from the fertilizer outlet was obtained by the sensor for every 0.2 s. Thirdly, a continuous 6 s with 30 mass values could be selected after the fertilizer was being discharged steadily, and Equation (4) could be used to calculate the CVFU. This testing method was equivalent to the distributor passing a 3 m discharging length at a steady forward speed of 0.5 m/s.

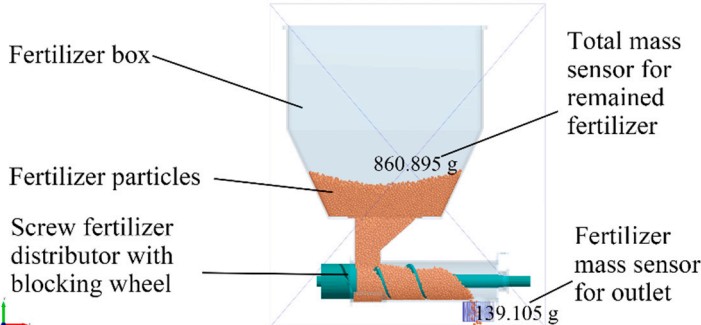

**Figure 3.** Simulation model and mass sensors.

### 2.5.3. Single Factor Tests Method

According to the design requirements of the horizontal screw conveyor, the relationship between screw pitch and its outer diameter was as follows:

$$P_t = k \cdot D \tag{5}$$

where $P_t$ is the pitch, mm; $D$ is the outer diameter of screw blade, mm; $k$ is the pitch coefficient, usually $k$ is 0.5–1.

Therefore, the range of $P_t$ was 22.5–45 mm calculated by Equation (5). As a result, the pitch was set to 25, 35 and 45 mm, respectively.

When the fertilizer distributor is in a static state, to ensure the fertilizer particles are fully accumulated in the fertilizer distributor, the outlet distance $S$ should be calculated as in Equation (6):

$$\begin{cases} S = r - P_t \\ r = \dfrac{D}{\sin \theta} \end{cases} \tag{6}$$

where $S$ is the outlet distance, mm; $P_t$ is the pitch, mm; $r$ is the stacking radius, mm; $D$ is the outer diameter of screw blade, mm; $\theta$ is the fertilizer stacking angle, °.

The stacking angle of commonly used fertilizer is about 30° [30]. If the stacking height is approximately equal to outer diameter $D$ = 45 mm, then the outlet distance $S$ should be about 33 mm to ensure the full accumulation of fertilizer particles under static conditions when the pitch $P_t$ is 45 m. Considering the influence on the fertilizer accumulation of the rotation of the screw shaft and the vibration of the fertilizer distributor, the outlet distance $S$ was set to a maximum value of 40 mm in the single simulation factor test.

Only one factor was changed for each test group, while the other factors were fixed, as shown in Table 2. The structure of the screw shaft with different number of screw heads is shown in Figure 4, and their pitch was 45 mm.

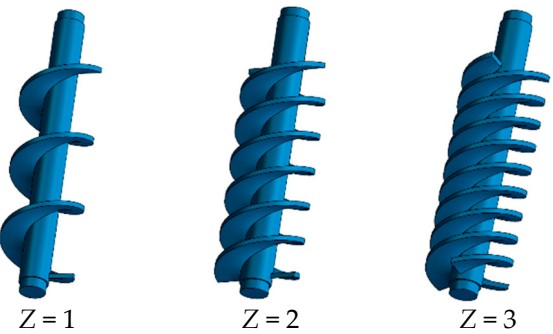

Z = 1       Z = 2       Z = 3

**Figure 4.** Screw shaft with different number of heads.

**Table 2.** Single factor simulation test table.

| Test Group | Inner Diameter of Screw Blade *d*/mm | Pitch of Screw $P_t$/mm | Number of Screw Heads *Z* | Outlet Distance *S*/mm | Blocking Wheel Opening Width *K*/mm |
|---|---|---|---|---|---|
| 1 | 17, 24, 31 | 45 | 1 | 0 | 30 |
| 2 | 17 | 25, 35, 45 | 1 | 0 | 30 |
| 3 | 17 | 45 | 1, 2, 3 | 0 | 30 |
| 4 | 17 | 45 | 1 | 0, 20, 40 | 30 |
| 5 | 17 | 45 | 1 | 40 | 10, 15, 20, 25, 30 |

### 2.5.4. Orthogonal Simulation Tests Method

To further clarify the effect of outlet distance *S*, pitch $P_t$, and the blocking wheel opening width *K* on fertilization uniformity of the screw fertilizer distributor, orthogonal simulation tests of the above three factors were carried out. In the tests, the outer diameter of screw blade *D* was set to 45 mm, the inner diameter of screw blade *d* was set to 17 mm, the number of screw heads was set 1, and the rotation speed of screw shaft *n* was set to 20 r/min. The influence factors of the orthogonal test are shown in Table 3.

**Table 3.** Factors and levels of influence of the orthogonal test.

| Level | Blocking Wheel Opening Width *K*/mm | Pitch of Screw $P_t$/mm | Outlet Distance *S*/mm |
|---|---|---|---|
| 1 | 30 | 25 | 0 |
| 2 | 20 | 35 | 20 |
| 3 | 10 | 45 | 40 |

The CVFU was used as an evaluating index. Table $L_9(3^4)$ was selected for the orthogonal test, and Minitab was used to deal with the results of the orthogonal test, test statistical assumptions, and variance analysis. The experimental arrangement and results are shown in Table 4.

**Table 4.** Orthogonal test results.

| Test NO. | Factors | | | CVFU *CV*/% |
|---|---|---|---|---|
| | *K*/mm | $P_t$/mm | *S*/mm | |
| 1 | 1 | 1 | 1 | 53.41 |
| 2 | 1 | 2 | 2 | 41.13 |
| 3 | 1 | 3 | 3 | 19.27 |
| 4 | 2 | 1 | 2 | 34.49 |
| 5 | 2 | 2 | 3 | 28.10 |
| 6 | 2 | 3 | 1 | 38.72 |
| 7 | 3 | 1 | 3 | 37.60 |
| 8 | 3 | 2 | 1 | 47.65 |
| 9 | 3 | 3 | 2 | 29.95 |
| $k_1$ | 37.94 | 41.83 | 46.59 | |
| $k_2$ | 33.77 | 38.96 | 35.19 | |
| $k_3$ | 38.40 | 29.31 | 28.32 | |
| *R* | 4.63 | 12.52 | 18.27 | |
| Order | | $S > P_t > K$ | | |

### 2.6. Method and Metarial of Bench Tests

#### 2.6.1. Test Bench and Fertilizer

According to the optimal structural parameters obtained from the simulation test results (*D* = 45 mm, *d* = 17 mm, $P_t$ = 45 mm, *S* = 40 mm), the FDM rapid prototyping technology was used to trial-produce the distributor, and the test bench shown in Figure 5

was built in the Engineering Training Base in Huazhong Agricultural University, China. The bench consisted of a blocking wheel-type screw fertilizer distributor, a driving moto (12V 60W), a moto driver (based on L298N), a photoelectric sensor (based on LM393 and H2010 Optocoupler), a Raspberry Pi 3B, 5V and 12V batteries.

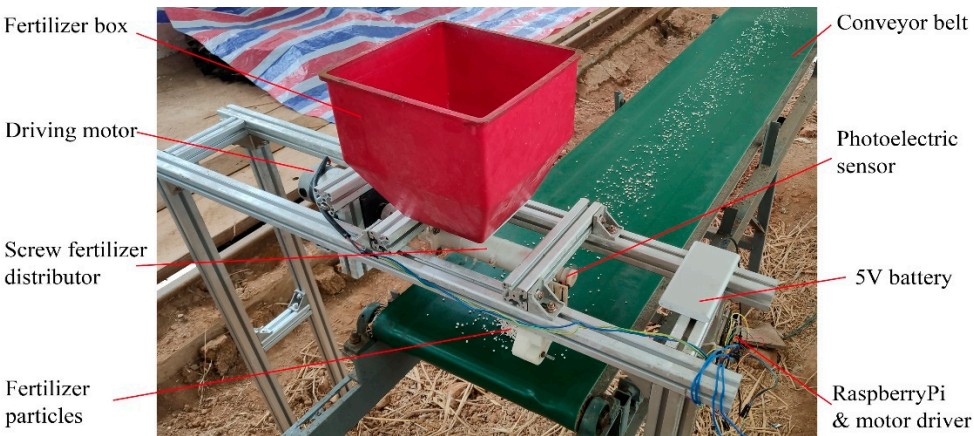

**Figure 5.** Testing bench.

The fertilizer used in bench test was Xiangyun high-sulfur-based compound fertilizer (N-$P_2O_5$-$K_2O$ content was 15%-15%-15%, bulk density was 1.20 g/cm$^3$ and moisture content was about 7.35%).

### 2.6.2. Verification Test

In order to verify the results of the simulation tests, the CVFU of the optimized distributor was tested under the same conditions of the simulation tests. In the test, the rotation speed of screw shaft was set at 20 r/min, while the forward speed of the conveyor belt was set to 0.5 m/s. The blocking wheel opening width $K$ was set to 10, 20, and 30 mm, respectively, and each test was repeated three times. After carrying out the test, the CVFUs of the fertilizer distributor were calculated according to the method in Section 2.4.2.

### 2.6.3. Performance Tests of the Screw Fertilizer Distributor

When evaluating the performance of the optimized screw fertilizer distributor, the CVFS and the FAPL were selected as the evaluating indexes, and blocking wheel opening width $K$ (which was set to 10, 20, 30 mm) and rotating speed of the screw shaft (which was set to 20, 30, 40, 50, 60 r/min) were taken as the influence factors.

In the test, the distributor was operated according to the above specified parameters first. After the fertilizer particles were discharged steadily, a collecting bucket was placed under the fertilizer outlet. When the fertilizer was collected for 2 min, the bucket was removed. Then the total mass of the collected fertilizer was weighted by an electronic balance. Each step was repeated for five times. After the tests, FAPL and CVFS were calculated according to the method in Sections 2.4.1 and 2.4.2, respectively.

### 2.7. Field Tests

Five screw fertilizer distributors were installed to a DAIDO DXZ830 rice seeding machine (10 rice seeding rows, with a working width of 2.55 m), as shown in Figure 6. Each screw fertilizer distributor corresponded to two rice seeding rows. The distributors were driven by a motor (12 V 100 W), and the motor speed controlling system was the same as the [31]. The speed of screw shaft could be synchronized with that of the (power-take-off) PTO at a set transmission ratio (PTO: screw) by the controlling system, which was set according to the target fertilizer amount of per unit area and the FAPL. In this research, it was set to 11.7:1 (calculated with a target fertilizer amount of per unit area of 400 kg/hm$^2$ and a FAPL of 39 g/r).

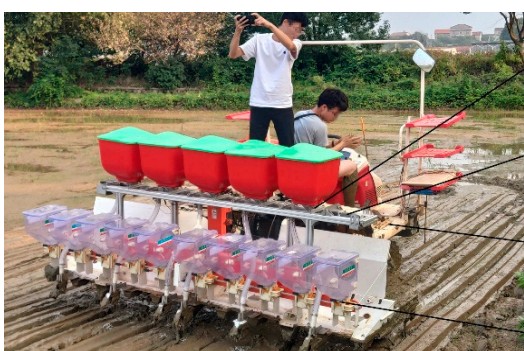

**Figure 6.** Screw fertilizer distributors with DAIDO DXZ830 rice seeding machine.

The fertilizer used in field test was Guihu compound fertilizer ($N$-$P_2O_5$-$K_2O$ content was 15%-15%-15%, bulk density was 1.11 g/cm$^3$, and moisture content was about 3.71%).

The paddy field was divided into six experiment plots, each plot had a length of 40 m and a width of 2.55 m. The field test was repeated three times, each test fertilized in two plots. After the test, the CVFS, variation coefficient of the fertilization consistency in different rows (CVODR), and the relative error of fertilization amount were measured.

The CVODR was also calculated by Equation (4), where $m_i$ is the total quantity of the *i*-th distributor during the test ($i = 1, 2, 3, 4, 5$), g; *m* is the average value of fertilizer quantity of the 5 distributors, g; *a* is the total number of the distributors; *S* is the standard deviation of fertilizer quantity of the 5 distributors, g; *V* is the CVODR, %.

The relative error of fertilization amount was calculated according to Equation (7)

$$\gamma_s = [10(W_0 - W_1)/A - T]/\text{T} \times 100 \tag{7}$$

where $\gamma_s$ is the relative error of fertilization amount, %; $W_0$ is the fertilizer amount added into fertilizer box before the test, g; $W_1$ is the fertilizer amount remaining in the fertilizer box after the test, g; *A* is the working area; m$^2$; *T* is the target fertilizer amount of per unit area, kg/hm$^2$.

## 3. Results

### 3.1. The Single Factor Tests

#### 3.1.1. Inner Diameter of Screw Blade

The simulation test results showed that the FAPL of the screw fertilizer distributor was 39.37, 32.86 and 24.05 g/r, and the CVFU was 36.65%, 40.44% and 38.77% respectively, for the three different inner diameters of 17, 24, and 31 mm. When other factors were fixed, the FAPL of the distributor decreased significantly, and CVFU did change within the error tolerance with the increase of the inner diameter of the screw blade.

#### 3.1.2. Pitch of the Screw

When the pitch was set to 25, 35, and 45 mm, the results showed that the FAPL was 20.08, 29.32, and 37.41 g/r and the CVFU was 53.41%, 47.27%, and 36.65%, respectively. The results indicated that by keeping the screw diameter, the outlet distance, the rotation speed, and the blocking wheel opening width constant and increasing pitch within the range of 25–45 mm, the FAPL of the screw fertilizer distributor increased, but the CVFU decreased. Therefore, when the outer diameter of the screw blade is 45 mm, the pitch should be set to 45 mm, which could improve the fertilization uniformity of the screw fertilizer distributor.

#### 3.1.3. Number of Screw Heads

The results showed that when the number of screw heads was 1, 2, and 3, respectively; the corresponding FAPL of fertilizer distributor was 39.37, 32.62 and 23.48 g/r, and the CVFU was 36.65%, 53.83%, and 62.45%. The number of screw heads had a significant effect

on both the FAPL and the CVFU. With the increase of the number of screw heads, the FAPL decreased while the CVFU increased. The results showed that the best number of screw heads was $Z = 1$ under the parameter setting of this study.

### 3.1.4. Outlet Distance

The simulation results showed that when the outlet distance $S$ was 0, 20, and 40 mm, respectively. The corresponding FAPL was 39.37, 38.34, 37.41 g/r, and the CVFU was 36.65%, 26.94%, and 19.27%, respectively. With the increase of the outlet distances, the change of FAPL was not obvious, but the fertilization uniformity was significantly improved. Therefore, the optimal fertilizer outlet distance is 40 mm.

### 3.1.5. Blocking Wheel Opening Width

Figure 7a showed the influence of the blocking wheel opening width $K$ on the FAPL within the range of the blocking wheel opening width $K$ of 10–30 mm. There was a good positive linear relationship between the FAPL and the blocking wheel opening width $K$. The regression equation was $q = 0.48K + 24.38$, and the coefficient of determination ($R^2$) was 0.9557.

Figure 7b showed that when the range of the blocking wheel opening width $K$ was set to 10–30 mm, the CVFU decreased with the increase of the blocking wheel opening width. However, the linear fitting degree was not very high, and the coefficient of determination ($R^2$) was 0.8932.

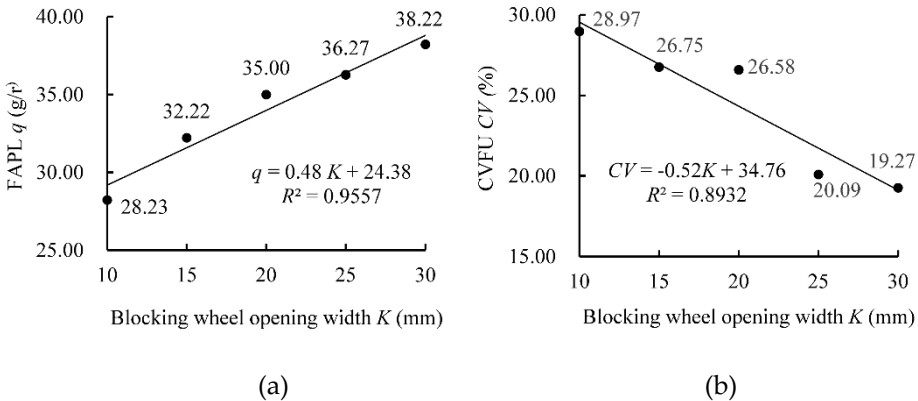

(a)                                    (b)

**Figure 7.** Effect of the blocking wheel opening width on the performance of fertilizer distributor. (**a**) Effect on fertilizer amount per lap (FAPL), (**b**) effect on coefficient of variation of fertilization uniformity (CVFU).

### 3.2. Orthogonal Simulation Test

The results of the orthogonal simulation test are shown in Table 4, the results of statistical assumptions for variance analysis (normal distribution of CVFU, homogeneity of variance, normal distribution of the residuals) are shown in Table 5, and the variance analysis results are shown in Table 6.

**Table 5.** Statistical assumptions for variance analysis.

| Item | | Test Method | *p*-Value |
|---|---|---|---|
| Normal distribution of *CV* | | Anderson-Darling test | 0.973 |
| | *K* and *CV* | | 0.344 |
| Homogeneity of variance | $P_t$ and CV | Bartlett's Test | 0.999 |
| | *S* and *CV* | | 0.828 |
| Normal distribution of the residuals | | Anderson-Darling test | 0.792 |

**Table 6.** Variance analysis performed on the orthogonal test results of CVFU (CV%).

| Source | SS | df | MS | F Value | Significant |
|--------|-----|-----|-----|---------|-------------|
| $P_t$ | 258.06 | 2 | 129.03 | 6.23 | * |
| $S$ | 510.98 | 2 | 255.49 | 12.34 | ** |
| $K \atop e$ $\Big\} e^\Delta$ | $39.01 \atop 43.80$ $\Big\} 82.81$ | $2 \atop 2$ $\Big\} 4$ | 19.51 | | |
| Sum | 851.85 | 8 | | | |

Critical value of *F*-test: $F_{0 \cdot 1}$ (2,4) = 4.32; $F_{0 \cdot 05}$ (2,4) = 6.94. * means the factor is significant when $\alpha$ = 0.1, ** means the factor is significant when $\alpha$ = 0.05.

The results of statistical assumptions for variance analysis showed that the *p* value of all the items was higher than 0.05. The data of CVFUs were normally distributed, the variances were homogenous between factors and CVFU (*K* and *CV*, $P_t$ and *CV*, *S* and *CV*), and the residuals were also normally distributed.

In the variance analysis, since $MS_K < MS_e$, when the blocking wheel opening width *K* was included in the error term, then the corrected error $e^\Delta$ was obtained.

Orthogonal simulation test and variance analysis results showed that the order of each factors on CVFU was ranked as: $S > P_t > K$. Among them, outlet distance *S* had an extremely significant effect on CVFU, the pitch $P_t$ had a significant effect on CVFU, and the blocking wheel opening width *K* had no significant effect.

The best combination of orthogonal test was $K_x P_{t3} S_3$ at a pitch $P_t$ of 45 mm and an outlet distance *S* of 40 mm. Combined with the single factor test results of the blocking wheel opening width *K*, when the blocking wheel opening width *K* was 30 mm at a rotating speed of 20 r/min, the CVFU was the smallest, which was 19.27%.

*3.3. Bench Tests*

3.3.1. Verification Tests

The results of verification tests showed that the corresponding CVFU was 29.43%, 20.40%, and 19.20% when the blocking wheel opening width *K* was 10, 20, and 30 mm, respectively. The CVFU decreased with the increase of the blocking wheel opening width *K*. Within the allowable error range, the simulation test results were consistent with the bench test results.

3.3.2. Performance Tests of the Screw Fertilizer Distributor

The performance testing results of the blocking wheel-type screw fertilizer distributor are shown in Tables 7 and 8 and Figure 8.

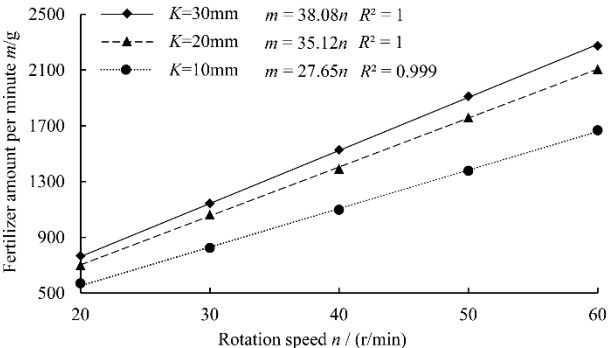

**Figure 8.** Effect of the rotation speed on the fertilizer amount per minute.

**Table 7.** Results of coefficient of variation of fertilization stability (CVFS) determination tests.

| CVFS *CV*/% | | Blocking Wheel Opening Width *K*/mm | | |
|---|---|---|---|---|
| | | 30 | 20 | 10 |
| Rotation speed *n*/(r/min) | 20 | 0.81 | 1.15 | 1.24 |
| | 30 | 0.76 | 0.85 | 0.84 |
| | 40 | 1.4 | 0.82 | 1.29 |
| | 50 | 0.67 | 1.72 | 0.49 |
| | 60 | 0.47 | 0.24 | 2.18 |

**Table 8.** Results of FAPL determination tests.

| FAPL *q*/g·r$^{-1}$ | | Blocking Wheel Opening Width *K*/mm | | |
|---|---|---|---|---|
| | | 30 | 20 | 10 |
| Rotation speed *n*/(r/min) | 20 | 38.33 | 35.03 | 28.49 |
| | 30 | 38.11 | 35.45 | 27.44 |
| | 40 | 38.19 | 34.81 | 27.43 |
| | 50 | 38.23 | 35.21 | 27.54 |
| | 60 | 37.90 | 35.13 | 27.79 |
| Average | | 38.15 | 35.13 | 27.74 |
| Variation coefficient of FAPL/% | | 0.42 | 0.67 | 1.60 |

The results of the CVFS determination tests showed that within the range of the blocking wheel opening width *K* of 10–30 mm and the rotating speed of screw shaft of 20–60 r/min, the maximum CVFS of the screw fertilizer distributor was only 2.18%.

The results of FAPL determination tests showed that the FAPL was not significantly affected by the rotating speed when the rotating speed of the screw shaft changed within the range of 20–60 r/min. This result indicated that there was a strict linear proportional relation between the fertilizer discharging rate and the speed of screw shaft.

In the range of 10–30 mm of the blocking wheel opening width *K*, the adjusting range of FAPL was 27.74–38.15 g/r and the FAPL increased with the increase of the blocking wheel opening width *K*. It realized the design goal of adjusting the FAPL by the blocking wheel.

*3.4. Field Test*

The results of field tests were shown in Table 9. It showed that the maximum CVODR was 3.13%, the maximum relative error of fertilization amount was 3.95%, and the maximum CVFS of five screw distributors was 2.36%. Figure 9 showed that the fertilizers were uniformly distributed in the fertilizer ditch by the optimized screw distributor.

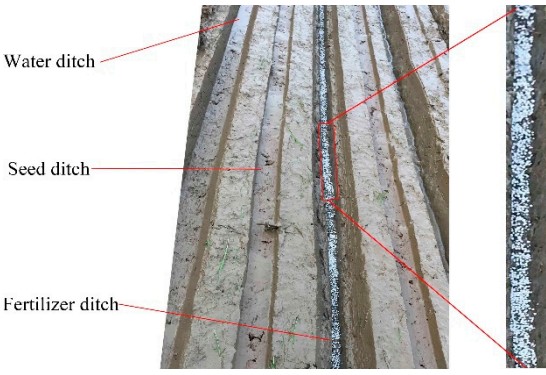

**Figure 9.** Presentation of paddy field after fertilization.

**Table 9.** Results of field tests.

| Test NO. | Fertilization Amount of Each Screw Distributor $(W_0 - W_1)$/kg | | | | | CVODR | Relative Error |
| | 1 | 2 | 3 | 4 | 5 | $CV$/% | $\gamma_{sl}$% |
|---|---|---|---|---|---|---|---|
| 1 | 1562.34 | 1587.45 | 1545.34 | 1530.98 | 1611.34 | 2.06 | 3.95 |
| 2 | 1624.15 | 1593.56 | 1538.64 | 1520.54 | 1580.12 | 2.66 | 3.71 |
| 3 | 1557.22 | 1650.97 | 1517.40 | 1555.22 | 1569.48 | 3.13 | 3.80 |
| CVFS $CV$ /% | 2.36 | 2.18 | 0.95 | 1.16 | 1.37 | | |

## 4. Discussion

### 4.1. Effect of Inner Diameter of Screw Blade on FAPL and CVFU

Figure 2 showed that when the fertilizer screw rotates, the change in the inner diameter of the screw has little effect on the stress of the fertilizer particles that have been filled into the distributer, and the filling rate will not change. The screw blade inner diameter has little impact on the CVFU, but the volume formed by the blade for each revolution of the screw will change and the volume of particles filled into the distributor will be changed, so the FAPL will be strongly affected by the screw inner diameter *d*.

It has been concluded that when designing a screw distributor, the inner diameter of the screw blade is a useful parameter to adjust the FAPL to the required value.

### 4.2. Effect of Pitch of Screw on FAPL and CVFU

According to Figure 2 and Equation (1), when the outer diameter of screw blade *D* was fixed, the lift angle of screw blade $\beta$ increased with the increase of screw pitch $P_t$, the friction angle $\alpha$ did not change, the axial component force $P_1$ decreased, and the circumferential component $P_2$ increased as well. Within a certain reasonable range, fertilizer particles moved along the axis and the of FAPL increased. However, the screw pitch $P_t$ increased to gravity *G* and the friction force $f_w$ generated by the outer shell wall of the fertilizer distributor was not enough to overcome the trend of fertilizer particles moving in a circle with the blade. In this situation, the efficiency of fertilizer particle axial transportation decreased gradually, until they were moving completely in the circumferential direction without axial movement. It is the reason why the pitch coefficient *k* was assigned a maximum value of 1 in Equation (5).

In addition, when the outer diameter *D* was fixed, increasing the pitch $P_t$ within a reasonable range, the angle between the horizontal plane and the cut plane of the screw blade decreased and the sliding speed of fertilizer particles from the screw surface decreased. At the same time the FAPL increased, so that in a rotating cycle, the time for fertilizer flowing out of the fertilizer outlet was prolonged and the time of no fertilizer discharging was reduced to improve the fertilization uniformity.

### 4.3. Effect of Number of Screw Heads on FAPL and CVFU

When the outer diameter *D*, inner diameter *d*, pitch $P_t$, outlet distance *S*, and the rotation speed *n* were fixed, with the increase of the number of screw heads *Z* there would be two main changes. On one hand, the axial force $P_1$ of screw blades on fertilizer particles did not change, but the circumferential force $P_2$ increased. On the other hand, the fertilizer transported in one cycle was discharged several times. All these changes would affect the CVFU of the fertilizer distributor.

In addition, with the increase of the number of screw heads *Z*, the total volume of screw blade increased and the proportion of fertilizer particles moving along the circumferential direction also increased, which led to a decrease of FAPL.

### 4.4. Effect of Outlet Distance on FAPL and CVFU

The outlet distance *S* does not affect the FAPL of a screw fertilizer distributor but affects the CVFU of it. When the outlet distance *S* is enough, fertilizer can be fully accumulated in

the buffer zone. The friction between the fertilizer particles and the shell of the fertilizer distributor, the screw shaft, and other particles will form a repose angle $\theta$, as shown in Figure 10. At this time, the fertilizer will move axially with the rotation of the screw shaft. When the overall transport distance of the fertilizer reaches $\delta$, the tangential component force $F_\sigma$ produced by the gravity $G$ of fertilizer particles in the shadow area outside the fertilizer pile will be greater than the friction force $f$, and the fertilizer within the scope of $\delta$ area will collapse freely from the bottom and flow out of the fertilizer outlet. The steadiness of the collapse will improve the CVFU.

If outlet distance $S$ is not enough, the accumulation of fertilizer in the buffer zone will be not sufficient, the unstable accumulation will be formed in the area outside the angle of $\theta$, and the fertilizer will slide unsteadily, which will reduce the fertilization uniformity.

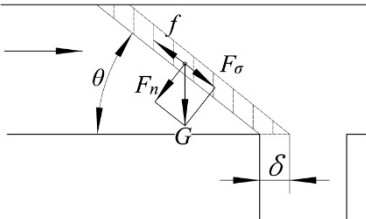

**Figure 10.** Effect of outlet distance on fertilizer amount uniformity.

### 4.5. Effect of Blocking Wheel Opening Width on FAPL and CVFU

When other parameters were kept constant, by reducing the blocking wheel opening width $K$ the space of the filling section of the screw fertilizer distributor became smaller, therefore the fertilizer filling rate $\varphi$ of the fertilizer conveying section was reduced. The FAPL of the screw fertilizer distributor $q$ decreased according to Equation (2).

With the increasing of the blocking wheel opening width $K$, the fertilizer filling rate $\varphi$ in the conveying section was increased, the fertilizer accumulated more fully, and the stability of fertilizer free collapsing was improved, so the fertilization uniformity was improved.

### 4.6. Effect of Rotation Speed of Screw Shaft on FAPL and CVFU

In theory, there is a critical speed $n_0$ of the screw shaft when the screw fertilizer distributor is working; that is, the highest speed at which the filling rate can maintain the maximum value when the fertilizer enters into the fertilizer distributor by gravity. If the rotation speed of the screw shaft is higher than $n_0$, the fertilizer filling rate will decrease with the increase of the rotation speed, which will lead to the decrease of the FAPL of the distributor. To ensure that the FAPL is less affected by the rotation speed, its working rotation speed should be less than the critical speed $n_0$. According to the structural characteristics of the screw fertilizer distributor, when the rotating speed is lower than the critical speed $n_0$, the higher the rotating speed, the higher the uniformity of fertilizer quantity. This research only optimized the fertilizer distributor at a low speed of 20 r/min to make the CVFU meet the requirements of the industry standard (NY/T 1003-2006).

### 5. Conclusions

(1) A blocking wheel-type screw fertilizer distributor was designed. Its working principle and the affecting force of fertilizer particles were explored in this research.

(2) Single factor and $L_9(3^3)$ orthogonal simulation tests based on EDEM software were carried out to optimize the parameters of the distributor at a low speed of 20 r/min. The results showed that when the outer diameter of the screw blade was set to $D = 45$ mm, the optimal parameters were: inner diameter of screw blade $d = 17$ mm, screw pitch $P_t = 45$ mm, outlet distance $S = 40$ mm, number of screw heads $Z = 1$, and the blocking wheel opening width $K = 30$ mm. The minimum CVFU value was 19.27% at the optimal parameters. The blocking wheel opening width $K$ had no significant effect on CVFU but had a significant effect on FAPL.

(3) According to the optimal structure parameters obtained from the simulation test, the FDM rapid prototyping technology was used to trial-product the fertilizer distributor, and the verification bench test was carried out under the same conditions as the simulation. The results showed that the structural parameters of the distributor were the same as those obtained by EDEM.

(4) Furthermore, bench and field tests were carried out to evaluate the performance of the distributor. The bench test results showed that the range of FAPL was 27.73–38.15 g/r when the blocking wheel opening width $K$ was 10–30 mm and the screw shaft rotation speed was 20–60 r/min. The FAPL was significantly affected by the blocking wheel opening width $K$, but less regulated by rotation speed of screw shaft. More importantly, the maximum CVFU and CVFS was 29.43% and 2.18%, respectively. The field test results showed that when setting the target fertilizer amount of per unit area to 400 kg/hm$^2$, the maximum CVODR was 3.13%, the maximum relative error of fertilization amount was 3.95%, and the maximum CVFS of five screw distributors was 2.36%. All these indexes are significantly better than the requirement in the "NY/T 1003–2006". The standard stipulated that the CVFU, CVFS, and CVODR should not be higher than 40%, 7.8% and 13%, respectively. This research solved the problems of the traditional screw fertilizer distributor only adjusting the fertilizer amount by rotating speed, and the fertilizer uniformity being poor at a low speed. It provided a meaningful reference for the optimization of the screw fertilizer distributor.

## 6. Patents

Two patents have been applied in China for the blocking wheel-type screw fertilizer distributor reported in this manuscript (Patent No. CN212436347U and Application No. CN111837549A).

**Author Contributions:** Conceptualization, X.Z. and G.Z.; methodology, X.Z.; software, X.Z.; validation, X.Z., G.Z., Y.H., A.E.S., J.F. and Y.Z.; formal analysis, X.Z.; investigation, X.Z., Y.H, A.E.S., J.F., and Y.Z.; resources, X.Z.; data curation, X.Z.; writing—original draft preparation, X.Z.; writing—review and editing, X.Z., G.Z.; visualization, X.Z.; supervision, G.Z.; project administration, G.Z.; funding acquisition, G.Z. All authors have read and agreed to the published version of the manuscript.

**Funding:** This work was financially supported by the Integrated Innovation of Mechanized Precision Fertilization Technology for Japonica Rice (Grant No. 2018YFD0301304-03) and Mechanization Technology Innovation of Ratoon Rice in the North of the Middle and Lower Reaches of the Yangtze River (Grant No. 2017YFD0301404-05), which are the sub-projects of the National Key Research and Development Project of China.

**Institutional Review Board Statement:** Not applicable.

**Data Availability Statement:** The data presented in this study are available on demand from the first author at (zhaxiantao@163.com).

**Acknowledgments:** This work was financially supported by the National Key Research and Development Program of China (Integrated Innovation of Mechanized Precision Fertilization Technology for Japonica Rice, 2018YFD0301304-03; Technological Innovation of Regenerated Rice Mechanization in the North of the Middle and Lower Reaches of the Yangtze River, 2017YFD0301404-05). We thank Mohamed Anwer for his linguistic assistance during the preparation of this manuscript. All supports and assistance are sincerely appreciated.

**Conflicts of Interest:** The authors declare no conflict of interest.

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
