# Peer review of "Structural Optimization and Performance Evaluation of Blocking Wheel-Type Screw Fertilizer Distributor"

_agriculture, doi:10.3390/agriculture11030248_

Round 1

Reviewer 1 Report

Dear Authors, first of all i want to congratulate for your hard work. Below you can find my comments.

I would suggest to add the sentences from line 92 to line 95 in the caption of figure 1

Line 167-167: In my opinion there were some mistypings..

2.4.2. Coefficient of variation of fertilization uniformity and stability (CVFUand CVFS)

CVFU and CVFS are important indices for assessing the uniformity and stability of…

Line 243

In my opinion the d in the equation (5) should be upper case

Paragraph 2.5.4 From line 267 to line 273

Have you checked the assumption of L9(34) orthogonal array?  As far as I know the additive assumption of orthogonal array implies that the individual or main effects of the independent variables on performance parameter are separable. Under this assumption, the effect of each factor can be linear, quadratic or of higher order, but the model assumes that there exists no cross-product effects (interactions) among the individual factors. That means the effect of a certain independent variable on performance parameter does not depend on the different level settings of any other independent variables and vice versa. If at anytime, this assumption is violated, then the additivity of the main effects does not hold, and the variables interact.

Line 274—277

Reading this part, I understood that you performed variance analysis on the CVFU used as an evaluating index for the orthogonal test..

Did you check the assumptions of Analysis of Variance? (normal distribution of the data, homogeneity of variance, normal distribution of the residuals?). Please indicate the test you performed to verify the assumptions and the relative results.

Line 373 the caption of the table 4 should be more informative

Table 4

I was not able to understand what 2.8refers to.

Test NO.                                 Factors                             CVFU CV/%

                        K/mm            Pt/mm            S/mm            2.8

1                        1                       1                        1             53.41

...                     ....                    ....                      .....

Line 374,  the caption of the table 5 should be more informative, i.e. “Variance analysis performed on the data CVFU (CV%)”.

Line 389-392

I am not so sure that you can state these sentences without a specific statistic analysis.

Line 452

In the pdf file the Figure 7 appears as a “completely white panel”. I was not able to visualize it.

Author Response

Please read the attachment.

Reviewer 2 Report

The paper presents important aspects regarding the distribution of solid chemical fertilizers

The experiment is well structured and described
However, I consider it necessary to present field tests, not just laboratory tests.
Also, the work does not result in maintaining the uniformity of distribution and the correlation with the working width of the fertilizing machines.

Author Response

Please read the attachment.

Round 2

Reviewer 1 Report

Dear Authors,
Once again, I would like to congratulate you on the work you have done. In my opinion, the improvements you have made to the manuscript are fully satisfactory.

Kind Regards.